# Scaling Ethereum 2.0s Cross-Shard Transactions with Refined Data Structures

Alexander Kudzin [1,*], Kentaroh Toyoda [2,3], Satoshi Takayama [1] and Atsushi Ishigame [1]

1 Department of Electrical and Electronic Systems Engineering, Osaka Metropolitan University, Sakai City 599-8531, Osaka Prefecture, Japan
2 Institute of High Performance Computing (IHPC), A*STAR, Singapore 138632, Singapore
3 Faculty of Science and Technology, Keio University, Yokohama City 223-8522, Kanagawa Prefecture, Japan
* Correspondence: alexander_kudzin@ieee.org

**Abstract:** (1) Background: To solve the blockchain scaling issue, sharding has been proposed; however, this approach has its own scaling issue: the cross-shard communication method. To resolve the cross-shard communication scaling issue, rollups have been proposed and are being investigated. However, they also have their own scaling limitations, in particular, the degree of compression they can apply to transactions (TXs) affecting how many TXs can be included in one block. (2) Methods: In this paper, we propose a series of novel data structures for the compiling of cross-shard TXs sent using rollups for both public and private Ethereum. Our proposal removes redundant fields, consolidates repeated fields, and compresses any remaining fields in the rollup, modifying its data structure to compress the address, gas, and value fields. (3) Results: We have shown that our proposals can accommodate more cross-shard TXs in a block by reducing the TX size by up to 65% and 97.6% compared to the state-of-the-art in public and private Ethereum, respectively. This compression in TX size results in an over $2\times$ increase in transactions per block (TPB) for our proposals targeting both types of Ethereum. (4) Conclusions: Our proposals will mitigate the scaling issue in a sharded blockchain that utilizes rollups for cross-shard communication. In particular, it will enable such sharded Ethereum networks to be deployed for large-scale decentralized systems.

**Keywords:** cross-shard transactions; sharding; rollup; scalability; Ethereum 2.0





## 1. Introduction

Blockchain technology has the ability to revolutionize decentralized systems, particularly those that use consensus to work towards a common goal, process data on the blockchain network in a decentralized fashion, and need to be highly secure and resilient. A good example of this is a distributed energy grid where users utilize a smart contract (SC) to transact energy in a trusted and efficient manner.

A major limiting factor in blockchains, such as Ethereum, as large-scale global systems such as Visa or an energy grid management system (EMS), is their low transactions per second (TPS) [1]. To resolve the scalability issue, sharding has been proposed [2,3]. Sharding uses multiple blockchains in parallel allowing for the processing of TXs locally on each shard after which the new shard states are synchronized via cross-shard transactions (TXs). Cross-shard TXs also occur when an account on one shard sends a TX to an account on another. By processing TXs in parallel, the TPS of the main blockchain is roughly multiplied by the number of shards minus the synchronization overhead. The use of sharding has moved the TPS bottleneck from the Blockchain to the cross-shard communication and synchronization method. This, in turn, has led to rollups being proposed as a way to reduce the number of and data size of cross-shard TXs. Rollups, as adopted by Ethereum 2.0, utilize a SC to compress and batch TXs reducing both the data size and the number of these messages, thus improving TPS. When a rollup SC receives a local valid TX with an address going to another shard, it begins the rollup process. TXs from the same source shard all

going to the same destination shard, e.g., from the $j$-th shard to the $j+1$-th shard, are batched into one rollup TX and signed by either the rollup SC or the miner who provided computation resources to the rollup [4]. Using rollups, scalability can be increased from hundreds of TPS to thousands of TPS [4,5]; however, hundreds of thousands of TPS are necessary for global systems such as an EMS. This limitation is of particular concern to the authors as it is a limiting factor preventing us from further exploring the potential of blockchain technology as the basis upon which a secure, resilient, and, decentralized EMS can be constructed and evaluated to determine its suitability for the new decentralized energy grid.

By examining rollups as proposed for Ethereum 2.0, we found that there is much room for improvement in how data is managed which, if restructured, would allow for an improvement in the level of compression for cross-shard TXs sent using rollups. In this paper, we propose a series of space-efficient data structures to improve the scalability of cross-shard TXs. We achieve this by compressing necessary fields and eliminating redundant fields. Our proposal is four-fold: the first two are targeted at public Ethereum, meaning that the proposals are generic. In contrast, the latter two leverage the characteristics of transactions as often seen in an EMS, making them more application-specific and better targeted at private Ethereum; however, there may still be certain use cases on public Ethereum, such as updating the coordinating layer, where they may be beneficial. We will clearly describe the two assumptions in Section 5.

Our contributions to research are;

- We propose a novel data structure for the Merkle tree used for address management in Ethereum, which we propose to combine with the Merkle tree used in rollups and the shard management lists to create a combined tree that allows for a reduction in field sizes.

  - We reduce the size of cross-shard TXs in rollups by up to 98% and 75% when compared to the conventional TX and state-of-the-art rollups.
  - We do this with minimal alterations to the rollup protocols as proposed for Ethereum 2.0 ensuring negligible system penalties.

- This reduction in TX size, for TX sent using our compression proposals for rollups, allows blocks to accommodate up to 98% and 65% more cross-shard TXs than when compared to the number of conventional TXs or state-of-the-art rollup TXs that can be sent per block [4].

The remainder of our paper is structured as follows: Section 3 introduces preliminaries. Section 2 summarizes related work. Section 4 describes the problem statement. Section 5 describes our methods. Section 6 shows the performance evaluation. Finally, Section 7 concludes the paper.

## 2. Related Work

In this section, we will review research on different scaling techniques applicable to Ethereum, each of which has a different area of focus [6,7]: (i) Changes to the protocol such as block size [8–10], latency, block interval [11,12], and gas costs [13,14]; (ii) using a different consensus algorithm such as Proof of Work (PoW), proof of share (PoS), and Proof of Authority (PoA) [15,16]; (iii) using side chains or channels such as lightning channels [17–19], and (iv) changes to the blockchain architecture such as sharding [3,20–22]. Most scaling approaches use a combination of these techniques to improve TPS [6].

Among the above approaches, sharding appears to offer a much higher potential throughput with fewer trade-offs and as such is becoming the preferred scaling approach [1,23]. However, sharding does have drawbacks that need to be overcome such as (i) attacks on single shards being potentially easier to execute than attacks on a blockchain, (ii) the processing overheads of running and maintaining the shards which could exponentially grow with more shards, and (iii) the cross-shard communication and synchronization methods. To overcome the limitations imposed by cross-shard communication, several dif-

ferent proposals have been made to create a secure and fast cross-shard protocol [21,24–26]. For instance, Kokoris-Kogias et al. used the concept of a two-phase commit from an atomic commit, but instead of applying it to the validators, they did so to the shards instead [24]. By locking the funds on each shard, they ensured that double spending did not occur and security was maintained. Nguyen et al. took the two-phase commit design of Kokoris-Kogias et al. and optimized it to minimize cross-shard TXs, such as only broadcasting TXs to the destination shard and placing TXs to minimize cross-shard components [25]. Dang et al. continued this line of research, further improving both the security and tamper resistance of the protocol through the use of both a two-phase locking and a two-phase commit protocol, which was run on a Byzantine tolerant shard [21]. This isolates the code from outside access during execution to prevent malicious blocking of the cross-shard TX. Liu et al. used a Byzantine fault-tolerant two-phase commit algorithm and Merkle trees to process multiple TX inputs together before then sending separate proofs [26]. This batching reduced calls to the Byzantine fault-tolerant algorithm, as only the Merkle root must be sent which speeds up TX processing. However, a large number of messages were still sent cross-shard in multiple rounds. The two-phase commit offers security at a low computational complexity; however, the number of messages is doubled, and message latency increases. Both of these drawbacks are due to the multiple commitment rounds.

To solve these issues of increasing message numbers, data size, latency, etc., various single-phase commitment, batching, and compressing techniques have been proposed [27–30]. In regards to TX batching, Das et al. [28] proposed the use of a single reference or coordinating shard with multiple worker shards. Worker shards process TX locally, before submitting them in sets to the coordinating shard where they are finalized using the consensus algorithm, after which sets of valid TXs are sent to their relevant destination worker shards and executed using data from the original worker shard to which the TX was submitted. As the consensus protocol only operates on the coordinating shard, worker shards can use a few processing nodes to seal TXs into sets for validation. This reduction in processing nodes combined with the batching of TX reduces intra-shard and cross-shard messages. However, TXs, intra-shard, and cross-shard are only validated after being processed by the consensus protocol on the coordinating layer, causing a potential bottleneck.

Some methods leverage polynomials. Li et al. utilize Lagrange-coded computing to enable distributed polynomial verification over a coded blockchain [27]. The results of verification can be obtained through decoding the Lagrange code, thereby using distributed computation to eliminate the need to use two-phase commit protocols for cross-shard TXs. However, all nodes must be fully aware of all TXs in the blockchain, requiring a significant increase in transmitted data. Wang and Raviv utilize a proof in the form of multivariate polynomials of a low degree [29], as opposed to the high degree polynomials used in Li et al. [27], thereby reducing difficulty growth with respect to the number of TX to logarithmic. To reduce cross-shard TX, a grid-style architecture for sharding, where TXs are distributed to array shards with senders along the *x*-axis and recipients along the *y*-axis, the outputs are batched and processed minimizing cross-shard TX. Finally, Wang et al. propose to use zones able to run consensus independently and in parallel to other zones with the aim of drastically reducing cross-shard messages [30]. This is because only batched and validated messages are sent between zones.

From these different approaches, we can draw some conclusions as to what can be done to improve the performance of the cross-shard communications method; (i) the number of commitment phases needs to be reduced to reduce the number of messages, (ii) the number of messages needs to be further reduced with TX batching, (iii) the data size of these messages needs to be reduced with compression, and (iv) messages/accounts need to be distributed in such a way that there is minimal need for cross-shard messages.

Rollups, as proposed for Ethereum 2.0, draw on all these separate areas of research to create a multi-faceted approach [4,5,31]: (i) A single-phase commit protocol based on polynomials is used to generate signatures. (ii) TXs are batched with common elements such as signatures and proofs shared by all TXs in the rollup. (iii) Data are compressed

with only the data relevant to cross-shard reconciliation included. (iv) TX and accounts are distributed to shards in a way that minimizes cross-shard TX.

In this way, rollups address the shortcomings of the above approaches whilst combining their best elements. However, rollups and the above approaches still have room for further improvement as we will show in the following sections.

## 3. Preliminaries

We explain the technological background of Ethereum 1.0 and its next generation, Ethereum 2.0, from the sharding and rollups perspectives. This is to better understand what problems our proposal is going to solve.

### 3.1. Ethereum 1.0

Ethereum is a digital currency that uses blockchain as its underlying network. It also supports SCs on the Ethereum virtual machine (EVM). An SC is a program to be executed on the EVM, allowing programs to operate on top of the blockchain to provide additional functionality. Computation on the EVM incurs a gas fee, which can be paid with Ethereum's currency, ETH.

There are two types of Ethereum. One is public Ethereum where anyone can transfer ETH or deploy and execute SCs by paying a gas fee. On public Ethereum 1.0, miners use the PoW consensus algorithm to process these transfers or SCs into blocks. The other one is private Ethereum which only limited users can manage and access. Private versions of Ethereum use permissioning certificates to regulate access. This can also be used as the basis of the consensus algorithm, PoA, which replaces miners with a few sealers to form transfers or SCs into blocks. As Ethereum is open-source software, anyone can deploy it on their own servers. This has led to many enterprises adopting Ethereum, altering protocols and parameters, and applying it to their specific use case. A few examples are asset tracking in a supply chain, utility management (e.g., electricity, gas, and water), and data management for multiple participants (e.g., medical records, identifiers, and financial information).

Data Structure of Conventional TX

Next, we explain how TXs on Ethereum are structured. Currency transfer and SC code execution are accomplished through a TX whose structure is shown in Table 1. Of the fields shown in Table 1, most can be of a variable length typically up to 32 bytes long. However, they also have a minimum size where only the value and data field can be 0 bytes in length. The other fields are at least 1 byte long except the gas limit field, which is at least 2 bytes long. These TX get collated together into a Block which has an upper limit to its size as such it is quite important to reduce the size of TXs, thereby enabling more TX per block, which results in an improvement in scalability. We will cover field size in detail in Section 6.

**Table 1.** Structure of a conventional Ethereum TX.

| TX FIELD | DESCRIPTION | FIELD LENGTH & OPCODE IN BYTES |
|---|---|---|
| CHAIN ID | Blockchain ID (e.g., 1 for the main Ethereum network) | 1 + 1 |
| RECIPIENT, SENDER | Recipient's and sender's addresses | 32 + 1 per address |
| VALUE | Value to be sent in Wei, the base unit of Ethereum where 1 ETH $= 10^{18}$ Wei | $\leq 32 + 1$ |
| GAS PRICE | Cost of a unit of computation as determined by Ethereum's gas tables in gWei where 1 ETH = $10^9$ gWei | $\leq 32 + 1$ |

**Table 1.** *Cont.*

| TX FIELD | DESCRIPTION | FIELD LENGTH & OPCODE IN BYTES |
|---|---|---|
| GAS PRIORITY FEE | These are priority fees much like tips which are paid to the miners | $\leq 32 + 1$ |
| GAS LIMIT | The maximum gas a user is willing to pay to process a TX | $\leq 32 + 1$ |
| NONCE | Counter for the number of TXs sent by the user | $\leq 32 + 1$ |
| ACCESS LIST | Optional list of accounts the TX will interact with | $32 + 1$ per address |
| SIGNATURE | The TX data are taken and signed using the user's private key | $68 + 1$ |
| V | A part of a signature used to prevent a replay attack | 1 |
| DATA | Used to deploy data or SC as well as to make a function call to a SC. Typically empty for currency transfer | $\leq$block size $+ 1$ |

## *3.2. Types of Merkle Trees*

In this section, we will explain how account information, addresses, balances, etc., are managed and dynamically updated in Ethereum using Merkle trees. To do this, we will first use Figure 1 to introduce the different types of trees.

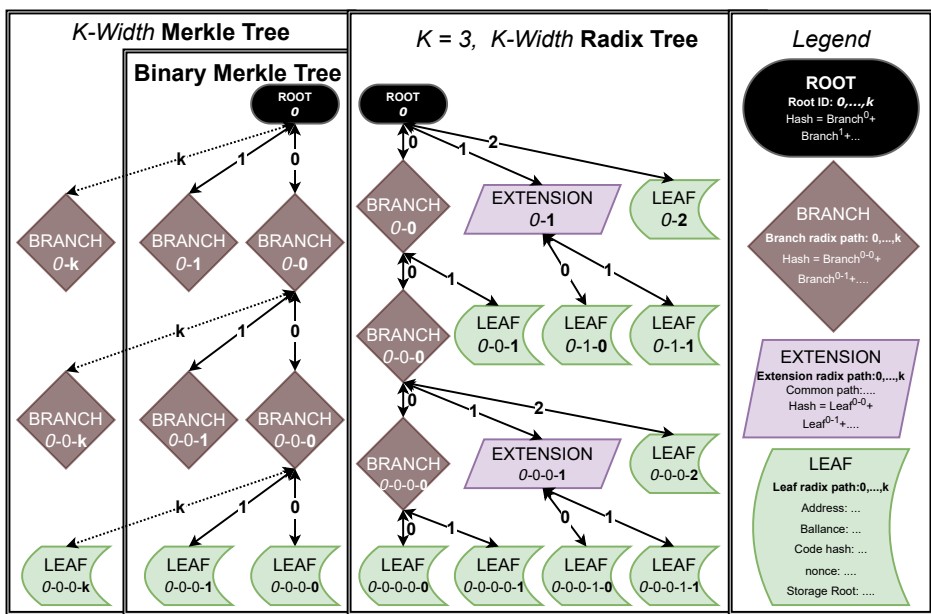

**Figure 1.** Comparison of binary Merkle, *k*-Merkle, and *k*-radix tree structures.

### 3.2.1. Merkle Trees

First, let us start by introducing a binary Merkle tree using the example in Figure 1 under the binary Merkle tree header. A Merkle tree is a binary data structure where every node has two children increasing from a single node at the top called the root to many nodes at the base, called leaves, with intermediary nodes called branches. The information to be stored in the Merkle tree is placed in the leaves. In this case, it is account information in the form of an address, a balance, a nonce, and storage/data hashes. Pairs of leaf nodes are then hashed together creating branches on a new layer that is half the width of the leaves on the layer below. These branches are intermediate nodes that just contain a hash whose length depends on the hashing algorithm, 256 bytes long for SHA-256. The hashing process is repeated until the highest layer has only one node, which is called the root. This root is a hash of all data in the Merkle tree; this is placed into the latest block to identify the current tree, allowing it to be securely dynamically updated. We can use the root of

a Merkle tree to quickly discover whether a given value has been tampered with. This is done by recreating the root by taking the hash of a given value, as well as all sister hashes, which are hashes along, but not on, the path to the root. We can then reprocess these hashes to create a root to see whether it matches the original root. When a branch has $k$ nodes, then it is called a $k$-Merkle tree. A binary Merkle tree is a specific case of $k$-Merkle tree with $k = 2$.

### 3.2.2. Radix Trees

Let us look at the radix tree as adopted by Ethereum and shown in Figure 1. Although radix trees are hashed the same way as Merkle trees, with lower nodes being hashed together to produce higher nodes, they have a different data structure built around the radix path which is a path from a parent to a child node that is numbered based on the width, $k$ value, of the radix tree. To store information in a radix tree, it is not placed at the lowest level; instead, the information is read as a radix path which is then followed until that path reaches a node on the edge of the radix tree, after which a new leaf node is created. In addition to the leaf, branch, and root nodes, radix trees in Ethereum also use the Extension node, which is a node containing a section of radix path spanning two or more nodes without any side branches that are common to two or more child nodes. These different node types and ways of storing information create a very different tree structure.

Now, let us use Ethereum to further explain how a radix tree is created. A 32-byte long Ethereum address is read in hex, where each hex value is read as a path in the $k = 16$-radix tree. A hex value ranging between 0 and 15 corresponds to a path from the current node to one of its children which are also ranging from 0 to 15. Once the last occupied node is passed, regardless of whether all hexes in the address have been read, a leaf node containing the account is created. If a leaf node is reached during this process, that leaf node is replaced with a branch node, and the account leaves are moved down a level. If a set of accounts have a common part of their address radix path where there are no other nodes off-shooting from the path, then an extension node is used instead of multiple branches where each branch would have had only one child before.

The main advantage of a radix tree is the ease of locating an account and its data from an address's radix path. A radix tree's disadvantage is that it is sparsely populated with leaves at varying heights until the radix tree is nearing full saturation. Therefore, if the lookup advantages of a radix tree could be transferred to a $k$-Merkle tree, say by the use of radix paths in all TXs, then a very dense data structure with leaves on a common level and fast look-up times could be created.

### 3.2.3. $k$-Merkle and Radix Trees

The process for creating a $k$-ary tree is similar to that of a Merkle or radix tree and results in similar structures. However, instead of $k = 2$ as for a binary Merkle or radix tree, a $k$-ary tree uses a higher value of $k$, as we explained in the previous section on how Ethereum uses $k = 16$ to create a hexadecimal $k$-radix tree.

We can see the examples of these $k$-Merkle and radix trees in Figure 1 where the increased $k$ children per node makes $k$ type trees much wider. This is an advantage, as there are fewer intermediary branch nodes and thus a shorter tree with a smaller data size, $1 + k^1 + k^2 + \cdots + k^n$ nodes as opposed to $2n - 1$ nodes, where $n$ is the number of addresses. For instance, for $n = 2^{40}$ addresses, a Merkle tree has $(2 \times 2^{40}) - 1$ nodes. A $k$-Merkle tree with $k = 256$ reduces the nodes in the tree by almost 50% to $1 + 2^8 + 2^{16} + 2^{24} + 2^{32} + 2^{40}$ nodes. The disadvantages of $k$-ary trees are that each node has a large number of sister nodes, $k - 1$, which increases the proof size when verifying a hash as an increased number of hashes must be sent. This is an increase from $O(\log_2 n)$ for a binary Merkle tree to $O(k \log_k n)$ for a $k$-Merkle tree which has a knock-on effect of increasing the cost of verification.

### 3.3. Ethereum 2.0

Ethereum 1.0 is currently part way through its transition to version 2.0 (The details of Ethereum 2.0 are not fully known at the time of writing and are subject to change. For the latest version of the Ethereum 2.0 code, see https://github.com/ethereum/consensus-specs/blob/v1.0.0/specs/phase0/beacon-chain.md, accessed on 10 September 2022) which is set to address its low TPS issue [22]. It has major technical differences when compared to Ethereum 1.0 such as the use of: (i) Sharding, which splits a blockchain into several inter-connected copies of itself which operate in parallel [22], (ii) Rollups, which compresses TXs who share the same source and destination shards as each other into one batch TX [4,5,31], and (iii) Replacing the consensus algorithm PoW with PoS, which is the staking of coins as collateral to attest to the validity of a created block [15]. This initial part of the upgrade was completed on 15 September 2022; however, as it is not directly pertinent, we will only cover sharding and rollups which are yet to be implemented on Ethereum 2.0.

#### 3.3.1. Sharding

Sharding is a technique to enable the parallel processing of multiple blockchains to improve scalability. An example of the sharding architecture proposed for Ethereum 2.0 can be seen here [22] and is explained in Figure 2. The beacon chain or coordinating layer coordinates and manages the shards by maintaining synchronization, ensuring a common ledger, providing shared secrets and random numbers, as well as managing the Merkle tree containing the account information. The shards receive sets of TXs from the mining pool. Under the Ethereum 2.0 proposal, these TX are split based on their transaction types. Specifically, the lower-numbered shards receive TXs for token transfer, whilst the higher-numbered shards receive TXs with data such as SCs, libraries, and other data from the message body. Miners then use an EVM to process the lower number shards' TXs and higher number shards' data into a block and update the Merkle tree's state on the coordinating layer. This necessitates cross-shard communication.

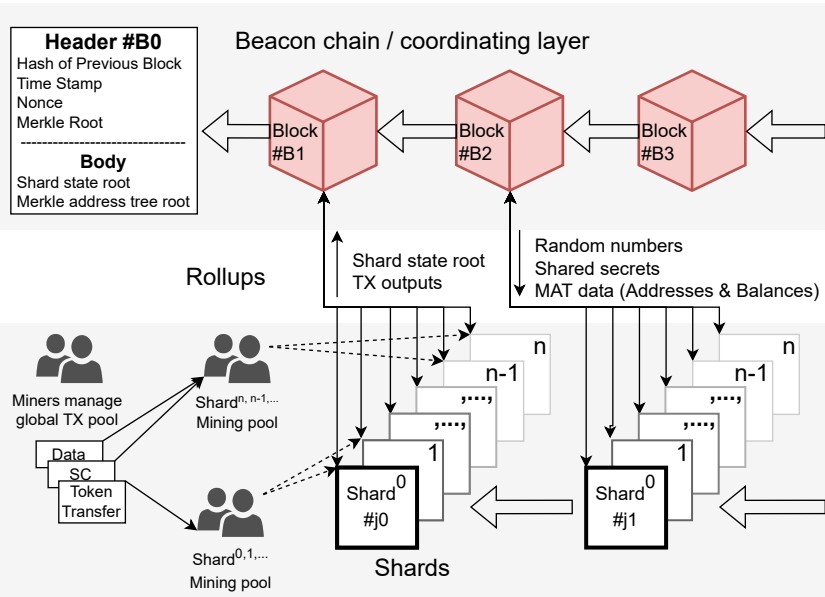

**Figure 2.** An illustration of sharding based on the current Ethereum 2.0 spec [22]. Note how rollups are used to provide cross-shard communication and update the coordinating layer.

The number of shards per epoch, where epoch is a set duration of time, is dynamically determined by the number of TXs to be processed [32]. The proposal for sharding states that it will start with 16 shards and gradually increase over time [22].

As with Ethereum 1.0, there is a dynamic radix-type Merkle tree to manage users' accounts, which, under the sharding proposal, remains on the main chain, the coordinating layer.

### 3.3.2. Rollups

If a TX created on a shard has an element on another shard, or when updating the Merkle tree on the coordinating layer at the end of an epoch, rollups are used. Rollups are SCs deployed on every shard, where they compress and batch TXs going across shards, which we call cross-shard TXs. We explain how a rollup is created using an example of Alice in the $j$-th shard sending some ETH to Bob in the $j + 1$-th shard. Alice creates a conventional style TX, as per Table 1, specifying Bob's address, shard, and account IDs (Bob's shard and account IDs can be determined by parsing the Merkle tree that manages accounts with Bob's address. Once Bob's account has been found, the radix path to Bob's account from the Merkle tree's root can be read, returning Bob's shard and account IDs), and the value to be sent to Bob placing this information into the `DATA` field. As per the message structure in Table 2, unnecessary and redundant fields are omitted or compressed. Specifically, the rollup SC's address and shard ID are then placed into the recipient address and shard fields. The total value to be sent is placed in the `VALUE` field; in this case, it is just the value being sent to Bob. All other fields are the same as for a conventional TX. Finally, after completing the TX, Alice hashes all the data before signing it using her private key. This TX is then sent to the local shards rollup SC to be validated and mined before it is formed into a rollup.

**Table 2.** Structure of a type of cross-shard TX called a rollup.

| TX FIELD | DESCRIPTION | FIELD LENGTH & OPCODE IN BYTES |
|---|---|---|
| HEADER | | |
| SHARD ID | The ID of the sending shard | 1 + 1 |
| ROLLUP SIZE | The size of the rollup in bytes | byte length + 1 |
| POST-STATE-ROOT | The Merkle root of the shard after the TXs | 4 + 1 |
| SIGNATURE | Signed by miners private key or rollup SC using a polynomial | 68 + 1 |
| BODY (repeated per TX) | | |
| SENDER RADIX PATH | The radix path from the Merkle root to the senders account leaf | 4 + 1 |
| RECIPIENT RADIX PATH | The radix path from the Merkle root to the recipients account leaf | 4 + 1 |
| VALUE | The value being sent is encoded as $x \cdot y \times 10^z$ | Up to 3 + 1 |
| GAS | Gas is replaced with a reference to a gas table | 1 + 1 |

Upon receipt of Alice's TX, her shard checks its validity. If successful, the funds are then locked on Alice's shard during the process of committing Alice's TX to the next block on the local $n$-th shard. Finally, her shard's state root is then committed to the coordinating layer. If it is not successful at any point, then Alice's funds are returned.

If successful, we can proceed to the next step, creating a rollup. TXs going from the $j$-th shard to the $j + 1$-th shard are placed into a mining pool for a rollup. Some fields are compressed and removed before being batched into a rollup. The structure of which can be seen in Table 2. We first explain the fields to be compressed. Specifically, there are six fields; (i and ii) the 32-byte sender and recipient addresses are replaced with 4-byte IDs based on the radix path to the accounts in the Merkle tree. (iii) The 32-byte value field is replaced with up to a 3 bytes long $x \cdot y \times 10^z$ format. For instance, 3 ETH instead of being sent as 3 Quintillion Wei, which is at least 8 bytes long, is sent as $x = 3, z = 19 \rightarrow 3 \times 10^{19}$, which is only 2 bytes long. (iv, v, and vi) The gas price, gas limit, and gas priority fields, each 32 bytes long, are replaced with a single 1 byte value referencing a common gas table. These compressed fields, namely the sender's ID, recipient's ID, value, and gas, are then batched together with other TXs from the same source shard going to the same destination

shard i.e., from the $j$-th to $j + 1$-th shard, and signed by the rollup SC. This signature is either completed using the private key of a miner or the SC itself.

By removing fields unnecessary for cross-shard reconciliation, compressing the remaining fields, and batching, we can typically reduce from 100 to 256 bytes to around 16 bytes whilst also reducing the number of TXs being sent across shards [5].

## 4. Problem Statement

Improving the specification of the current proposal for rollups, Ethereum's chosen cross-shard communication method, is important as Ethereum is upgrading to a sharded architecture with Ethereum 2.0 whose performance will be heavily dependent on the performance of the chosen cross-shard communication method. We argue that the performance of this cross-shard communication method, rollups, as proposed for Ethereum 2.0, can be much improved. Specifically, the data size of the address field in rollups can be much reduced if the $k$-radix tree responsible for managing addresses in Ethereum were restructured to a $k$-Merkle tree. This $k$-Merkle tree address management tree would also replace the Merkle tree used in rollups consolidating the roles of both trees into one. These changes to the tree type allow us to make the sharding committee a subset of the densely packed, uniform leaf height, $k$-Merkle tree. This then allows us to modify the way that the radix address is generated and read splitting it into a path to the subset from the root and a path within the subset to the leaf. Finally, we also found that we can compress the fields related to gas and values to further reduce the data size of rollups. This compressed data size would then allow more TXs to be included in one block, effectively improving scalability.

## 5. Proposed Method

We propose a series of compression techniques to reduce the data size of the TXs that comprise the body of the cross-shard type TX, rollups, the original structure of which is shown in Table 2. Here, we are proposing four compressions, each compression using a part of the last compression to create a greater TX data size compression. These four parts are as follows:

(i)     Address field compression (AC), we use our proposed Merkle tree structure to create commonality in the radix path used as an ID allowing the common path to be removed from the TX body;

(ii)    Address and gas field compression (AGC), we propose changes to when and how gas is validated to remove it from the TX body;

(iii)   Address, gas, and value field compression (AGVC), we propose changes to the rollup protocol, to instead of recording TXs, record the value change per user;

(iv)    Gas and value field compression without address field (GVCRO), we propose further changes to enable the elimination of the address field.

Of these four compressions, the first two proposals are applicable to both public and private Ethereum, and the third and fourth proposals have limited applicability to public Ethereum as they use system assumptions more common to private than public Ethereum. Now, let us explain each approach in order starting with our proposal for compressing and splitting the radix path used as the address in rollups.

### 5.1. Address Field Compression

Our first proposal is to compress the address field which we can apply to both public and private Ethereum. The core of our idea is to restructure the nodes in the $k$-ary tree to create commonality in the radix path IDs of all accounts on a shard, thereby enabling the movement of the common part of the path from the rollup body to the rollup header. This requires several steps: (i) We restructure the $k$-ary tree from radix, which has a fixed structure, to Merkle, whose structure can be updated. (ii) Next, we can restructure the nodes in the Merkle tree to create subsets/subtrees. (iii) This allows us to assign a sharding committee's account leaves to a subset/subtree. (iv) We have now created a string of nodes along the radix path that are common to all account leaves, where the lowest node along

this radix path becomes the shard ID in our proposal. (v) We place this shard ID into the rollup header with the opcode for either sender or recipient. (vi) We will also need to add a root ID to the shard ID after the opcode as, when the Merkle tree's structure changes at the end of an epoch, with the addition of new accounts and any shuffling/optimization, so will the radix paths. (vii) Finally, as all radix paths are prefaced with an opcode for either sender or receiver, the same one used in the shard IDs, we can remove the common, repeated, part of the radix path from all the account leaves, reducing the total rollup size.

The current radix path used in rollups under Ethereum 2.0 is the whole path, from the Merkle trees' root, through all the branches all the way to the final leaf, for both sender and receiver. This is akin to giving the full street address for both sender and receiver on the letter's envelope despite both parties being in the same company, just at different branches. However, what if we could group letters/messages going to the same branch office using a much shorter internal code? We could liken a shard to a branch office of a multinational company. In this case, the full street address of that branch office would be like the full address. This needs to include full address stems from shards in Ethereum 2.0 processing a number of TXs in the order they arrive, recording which accounts' TXs' are being processed on which shard using a list. This is much like if every branch office randomly received many messages from different remote users based on when the user sent their message. This random nature makes grouping users by branch very difficult; as a result, a full address must be given to each user, for each user involved.

In order to replace these full addresses with internal IDs, let us first make a set of users who are registered at a branch office, and place them together in one part of the Merkle tree used to manage user information. Let us also ensure that the tree structure is as densely packed as possible, that is, a high ratio of leaves to nodes, by using a $k$-Merkle tree.

This change in the way users are organized is the core of our first proposal to compress the address. If we change the way accounts are recorded and allocated to shards, from using a list to using a sub-section of the Merkle tree, the same tree that is used for managing users' accounts, we can order it into sets with a fixed address for that epoch. The blockchains' protocol needs to state how that list is ordered, created, and updated. It could be by the order of message arrival in an epoch as proposed for Ethereum 2.0 [4].

Next, we must structure the users from this fixed set forming them into a branch office by creating a Merkle root. This root needs to be created in such a way as to minimize the number of intermediate nodes. To do this, we set $k$ to be the number of users in the branch office. Alternatively, we can create departments inside the branch office if we want to use a smaller value of $k$. After structuring the users into a branch office, with or without departments, we can place the branch office into the company structure below the main office. If there are more branch companies than $k$, we will need to divide them by region by adding another layer to the Merkle tree. This gives us a Merkle tree with three main layers: the main office at the root, branch offices in the middle, and users in the leaves at the base. Each has a fixed location in relation to the other, and each layer is a subset of the layer above. We can now use this structure as a base for our internal address system.

Now that we have the company address structure, let us use it in place of the street address. Let us imagine a box, into which we will place that branch office's set of users' messages before sending it from branch office $n$ to branch office $n + 1$. For the internal post team, the messages inside the box would only need to give the internal address of the user, not the ID for their branch office or company, greatly reducing the message length, and allowing more messages to fit into the box. Obviously, branch offices $j$ and $j + 1$ in this example are analogous to the $j$-th and $j + 1$-th branches in the Merkle tree from which the $j$-th and $j + 1$-th shards in Ethereum receive their IDs.

Via this simplification, we have explained the overview of our proposed method to compress the ID further. Now, let us cover the technical aspects of this proposal.

### 5.1.1. Merkle Tree Structure

Figure 3 shows how a user's ID is structured and mapped in a $k$-Merkle tree. When the committee size, $c$, is less than or equal to $k$, the width of the Merkle tree, and the number of accounts to be stored in the $k$-Merkle tree, $n$, can be divided by $k$, such that $c \leq k$ & $\frac{n}{k} \leq k$, the full radix path for the User's account is formed by only three parts, namely the:

(i) Root ID, in the format, $i$ mod 256, is used to identify the $k$-Merkle tree across recent epochs, allowing the $k$-Merkle tree to be dynamically updated (Merkle trees outside this range can be identified by prefacing the ID with the coordinating layers' block number to which the Merkle tree to be referenced has been committed.),

(ii) Shard's branch ID, which is the lowest branch node common to all of the $c$ wide sharding committee, and

(iii) Account IDs for the $n$ accounts in the leaves.

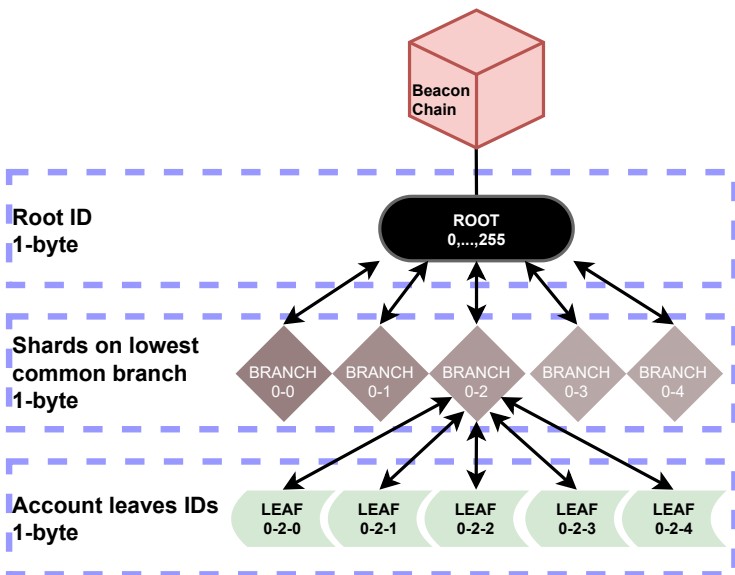

**Figure 3.** A $k = 5$ $k$-Merkle tree. The 1-byte Merkle root ID allows the correct Merkle tree to be easily identified across up to 256 epochs allowing the tree to be updated/shuffled each epoch. Shards are placed on the lowest branch common to the sharding committee of five account leaves.

In the case that an implementer of our proposal wishes to use a committee size, $c$, or $k$ value for the $k$-Merkle treewidth not within our proposed range where $k$ and $c \leq 256$, we have the following two exceptions. The first exception is that where we set a larger committee, $c$, than $k$-Merkle treewidth, $k$. If $c \nleq k$, then there will be intermediate nodes between a shard branch ID and account ID. This will necessitate the addition of intermediate shards to the Merkle tree and subheadings to the rollup in order to prevent an increase in the size of the account ID from 1 byte. If this is not implemented, then the performance of the proposals will be greatly impacted. Even if exception 1 is implemented the outcome will be sub-optimal, as there will be a slight performance penalty whose severity depends on the ratio of additional subheadings to messages in the rollup body.

The second exception is when we set more shards than the $k$-Merkle treewidth, $k$. If $\frac{n}{k} \nleq k$ but $\frac{n}{k} \leq k^j$, then there will be intermediate nodes between a root ID and shards' branch ID. The number of layers of intermediate nodes will be $j$. This exception does not impact the proposal.

The optimal tree is designed such that $c \leq k$ & $\frac{n}{k} \leq k$, which will result in the ID being only composed of one part, which, ideally, will only be 1 byte, hex or bit long.

Note that we include a root ID, which is not typically included in radix paths, in order to meet the requirement for a dynamic, updateable, tree. As such, the radix path is prefaced with a root ID. This allows the $k$-Merkle tree to be shuffled and updated at the end of each

epoch without invalidating an address. Outside of updating the tree to reflect new balances and accounts, the shuffling of the *k*-Merkle tree may be necessary either for optimization reasons or to prevent attacks by using a protocol like Cuckoo [33].

Finally, as mentioned earlier, we use a *k*-Merkle tree instead of a radix tree, as radix trees are largely empty with nodes at different heights until they are nearly saturated. As such, the IDs produced would be much longer and not offer the same level of compression.

### 5.1.2. Compressed ID Structure

Using the above Merkle tree structure, we can see that our proposal has the shard ID form part of the full radix path to each account. Therefore, let us split the full radix path into two parts at this point, namely (i) a part common to all senders or all recipients participating in a rollup that is comprised of the root ID and shard ID and (ii) a part unique to only the participant, the account ID. We will call these users who are participating in a rollup participants from here on.

We can distinguish a participant's role by using an opcode as an identifier, allowing us to link the sending or recipient shard ID in the rollup header with the multiple messages in the body. Thus, by changing the way that accounts on shards are managed and recorded, we can then change the Merkle tree structure, which enables us to change the ID structure. When repurposing the existing opcodes, we can reduce the size of each account ID in the message body from 4 bytes to just 1 byte per account by replacing the arbitrary shard number with our 2-byte shard ID in the header, as the shard ID is common to all senders or all recipients. These changes result in the compressed message format shown in Table 3.

**Table 3.** Structure of a cross-shard TX with our proposed address compression.

| TX FIELD | BYTES | OPCODE IN BYTES |
| --- | --- | --- |
| HEADER | | |
| SHARD ID | 1 byte (root ID) + 1 byte (shard ID) for up to 256 shards per layer & 1 byte per layer in the Merkle tree | 1 |
| ROLLUP SIZE | 1 byte per 256 bytes in rollup | 1 |
| POST-STATE-ROOT | 4 | 1 |
| SIGNATURE | 68 | 1 |
| BODY | | |
| SENDER ACCOUNT ID | 1 | 1 |
| RECIPIENT ACCOUNT ID | 1 | 1 |
| VALUE | Typically 2 or 3 | 1 |
| GAS | 1 | 1 |

### 5.1.3. Address Compression Procedure

The procedure for creating a rollup is largely the same as the process described in Section 3.3.2 except for the parts modified in our algorithms. Algorithm 1 shows our address compression method. It takes as an input valid TXs that have been mined on the local shard and outputs a radix path cut into two parts, a shard ID and an account ID, as per our proposed format. Algorithm 1 does this by taking the addresses from the input TXs and parsing the Merkle tree that manages accounts to locate the accounts named in the TX. After this, it returns the full radix path cut into two parts, the shard ID and the account ID. Algorithm 1 also performs a validity check by doing this process again in reverse, this time with the full radix path. This enables it to check that the address at the radix path and radix path at the address match. Previously, the shard and account IDs needed to be found by parsing separate lists: one for shard IDs and another for which users are assigned to those shards.

---

**Algorithm 1** Address compression.

    **Input**: $\mathcal{T}$, valid TXs that have arrived at shard$_j$, & Global Merkle address tree state
    **Output**: Shard and account IDs with opcodes for senders(s) and recipient(s)

1: **for** $TX \in \mathcal{T}$ **do**
2:    **for** sender and recipient in $TX$ **do**
3:                 ▷From $TX$ get all sender/recipient radix paths and addresses.
4:       from the global Merkle address tree get
5:       $R \leftarrow$ radix paths from $TX$
6:       $A \leftarrow$ addresses from $TX$
7:       **if** $A, R ==$ radix && $R, A ==$ address **then**
8:          ▷Verify the radix path and address match using the global Merkle address tree
9:          $h \leftarrow$ Shard height in the global Merkle address tree
10:           ▷Using the radix path, get the shard's height: $h$ from Merkle address tree
11:         Shard ID $\leftarrow R[: h]$
12:         Account ID $\leftarrow R[h :]$
13:      ▷Assign the opcodes for sender/recipient from the EVM to maintain ID integrity
14:         Sender shard ID $\leftarrow$ Sender opcode
15:         Sender account ID $\leftarrow$ Sender opcode
16:         Recipient shard $\leftarrow$ Recipient opcode
17:         Recipient account ID $\leftarrow$ Recipient opcode
18:       **end if**
19:    **end for**
20: **end for**
21: **return** Shard and Account IDs with opcodes for Sender(s) and Recipient(s)

---

AC, Section 5.1.2, informs Algorithm 1 which describes the procedure for generating the shard and account IDs by splitting the user's radix path at the shard height. It also describes an additional step in the validation process, after an ID and address lengths are checked in line with current practice, Algorithm 1 introduces a check to ensure that the address at the ID and ID at the address match. After these checks, the shard ID can then be placed into the rollup header, replacing the previous arbitrary shard ID. Similarly, the account ID can then be placed into the rollup body replacing the previous full radix path. However, now, in order to read the full radix path, the longer shard ID and shorter account ID must be read together.

*5.2. Gas Compression*

Our proposal for gas compression is intended to but is not limited to being used in combination with AC. Reviewing Table 3, we can see that each message has its own gas. However, as part of the Ethereum 2.0 spec, every action on the Ethereum network will have a base gas fee, and any additional gas, called priority, is just a fee to prioritize the TX. Therefore, we know the gas cost per byte to verify a rollup when creating the rollup or the source TX.

Next, we must consider how rollups are verified. When a rollup is received, its signature is validated, which is created from all the TX in the rollup body verifying them all at the same time. The cost of this batch verification is then split across the TX.

To avoid any TX having insufficient gas, or a TX over or underpaying any priority, and to also simplify this process, let us also make gas per rollup and not per TX. We shall also evaluate gas on the sending shard by checking that the gas in a source TX is greater than the three base fees; (i) source TX validation, (ii) rollup creation, and (iii) rollup validation. To ensure no one over or underpays, we can only include in the rollup any priority fee up to the minimum yet unspent priority fee included in a participant source TX. However, it can be debated as to whether any priority fee is necessary as the high density of gas in a rollup makes them very appealing to miners. Finally, let us use the field rollup byte length to allow us to convert gas from total gas per rollup to gas per byte, as follows: `Total gas`/`Rollup length` $=$ `Gas per byte`. By doing so, we can compress the gas field to a value that can be represented by one byte.

The miner on the receiving shard can then validate that the gas is sufficient in its compressed form and that the signature is correct, which will also validate the TX—after which the miner will receive any unburnt gas as a reward, as follows:

$$(\texttt{Rollup length} \times \texttt{Gas per byte}) - \texttt{base fee} = \texttt{miners reward}. \tag{1}$$

Applying this proposal for common gas to our proposal from AC, we obtain the following Table 4. These two changes have combined to compress the average TX from approximately 12 bytes for state-of-the-art rollups to approximately 6 bytes depending on the number of TX in the rollup. It is not exactly 12 or 6 bytes as the data size of the header must be divided between each TX in the body.

**Table 4.** The data format for a rollup using our proposed address and gas compression.

| TX FIELD | BYTES | OPCODE IN BYTES |
|---|---|---|
| HEADER | | |
| SHARD ID | 1 byte + 1 byte for up to 256 shards per layer & 1 byte per layers in Merkle tree | 1 |
| ROLLUP SIZE | 1 byte per 256 bytes in rollup | 1 |
| GAS | 1 | 1 |
| POST-STATE-ROOT | 4 | 1 |
| SIGNATURE | 68 | 1 |
| BODY | | |
| SENDER ACCOUNT ID | 1 | 1 |
| RECIPIENT ACCOUNT ID | 1 | 1 |
| VALUE | typically 2 or 3 | 1 |

Gas Compression Procedure

As gas is a dynamic value that is given in payment to the miner who validates the rollup, it is not possible to completely remove it. However, using Ethereum 2.0s, gas tables, which give a gas cost per action at a point in time, it is possible to pre-validate the amount of gas necessary. Using the Ethereum 2.0s gas tables and the field rollup length, we can compress the total gas for all participating TX to gas per byte as we show in Algorithm 2.

Algorithm 2 takes as an input validated TXs and their IDs, in line with the procedure described in Section 5.2. Gas is then accumulated into a single variable before; in the penultimate step of creating a rollup, it is divided by the rollup length, after which the rollup can be signed and dispatched to the rollup to the recipient shard.

*5.3. Address and Gas Compression with Value Batching*

Our second proposal builds on the previous compressions and is intended to be used to compress the size and number of TXs when multiple TXs per user per epoch are common. The core assumption of multiple TXs per user per epoch being common makes this proposal more applicable, but not limited, to private Ethereum. On public Ethereum, sets of batch TXs, or, frequent trades between a set of users on different shards, or, updating the coordinating layer at the end of an epoch may fulfill this assumption. However, the target use cases where it would be frequently used include, but are not limited to, an energy management system where a user transacts with multiple other users each epoch to buy and sell energy to meet their requirements.

---

**Algorithm 2** Gas compression.

---

   **Input**: The output of Algorithm 1, Valid TXs that have arrived at shard$_j$ and have had their shard and account IDs generated.

   **Output**: Rollups with only one gas field, denoted in gas per byte, in the header.

1: **if** Recipient shard ID in rollup pool **then** ▷Update an existing rollup header with the new TX elements

2:     `Rollup total gas += TX gas`                                                    ▷Add gas to gas in header

3:     `Rollup length += TX length`                          ▷Increase the rollup length by the TX length

4: **else if** Recipient shard ID not in rollup pool **then**   ▷Create a new rollup and place the TX elements into the rollup header.

5:     `Rollup header ←` $\begin{cases} \texttt{Sender shard ID and opcode.} \\ \texttt{Recipient shard ID and opcode.} \\ \texttt{Gas and opcode.} \\ \texttt{TX length and opcode.} \end{cases}$

6: **end if**                                                              ▷Add a TX's elements into the rollup body.

7: `Rollup body ←` $\begin{cases} \texttt{Sender account ID and the sender's opcode,} \\ \texttt{Recipient account ID and the recipient's opcode,} \\ \texttt{Value and value opcode} \end{cases}$

8: Generate the post-state root ▷Now that the rollup is complete, generate and replace the dummy post state root and signature in the header with the completed items:

9: `Rollup ←` $\begin{cases} \texttt{Post-state root and opcode,} \\ \texttt{Updated rollup length and opcode,} \end{cases}$

10: `Rollup gas ←` `Rollup total gas/Rollup length` and opcode        ▷Calculate gas per byte, replacing the 1 byte placeholder in the header

11: **if** `len(Rollup gas)` $\neq 1$ **then**

12:     `Rollup length ← len(Rollup)` and opcode

13:     `Rollup gas ←` `Rollup total gas/Rollup length` and opcode

14: **end if**

15: `Generate Signature`

16: `Rollup ← Signature and opcode`

17: **return** Rollup

---

For this compression, and further sections, we will also need to define a couple of key terms: (i) A user is a holder of an account that is included in the Merkle tree responsible for account management. (ii) A participant is used to denote a subset of users who are currently participating in a rollup or other TX type. We use this to differentiate between users in a sharding committee and users participating in a rollup from that shard to another.

Now, let us start by explaining the main idea behind this compression. A potential use case for a rollup is to update a user's net balance change at the end of an epoch. For example, Alice sends Bob and Charlie five tokens each. Bob also sends Alice and Charlie five tokens each. However, Charlie is on another shard, so, to avoid sending multiple costly messages, Alice and Bob choose to send a rollup. Therefore, the rollup SC would see a total value change for each participant as follows: Alice –5 tokens, Bob –5 tokens, and Charlie +10 tokens. It is through this value batching process that we reduce the number of TX from four to three. Based on this, let us change the structure of a rollup replacing TXs and their pair of users, sender, and recipient, with a net value change per participant. We illustrate this new structure in Table 5.

**Table 5.** The data format for a rollup using our proposed address, gas, and value compression.

| TX FIELD | BYTES | OPCODE IN BYTES |
|---|---|---|
| HEADER | | |
| SHARD ID | 1 byte for up to 256 shards per layer & 1 byte per layers in Merkle tree | 1 |
| ROLLUP SIZE | 1 byte per 256 bytes in rollup | 1 |
| GAS | 1 | 1 |
| POST-STATE-ROOT | 4 | 1 |
| SIGNATURE | 68 | 1 |
| BODY | | |
| PARTICIPANT RADIX PATH | 1 | 1 |
| VALUE | typically 2 or 3 | 1 |

Going from a structure like that of Table 2 with two participants per TX, sender, and receiver, resulting in a total TX size without opcodes of 12 bytes, comprising: 4 bytes for each sender and recipient, 3 bytes for a value, and 1 byte for gas.

To the structure in Table 5 with two single participant state changes, each being 3 to 4 bytes without opcodes, or 5 to 6 with opcodes, totaling between 6 and 12 bytes for both state change TX. If we were just to apply value batching to state-of-the-art rollups then each state change would be comprised of a 4 byte account ID field, a 2 to 3 bytes long value field, and a 1 byte gas field, with each field having its' own 1 byte opcodes.

Alternatively, we can use the TX structure from AC in Table 3 and common gas, from Section 5.2, in combination with this value compression to further compress the two participants' addresses, resulting in each state change TX totaling 3 to 4 bytes without opcodes, comprising a 1 byte account ID and 2 to 3 bytes value field.

We can see that, by just applying this value compression, we do not achieve a reduction in data size if the average number of TXs per participant is too low. If we assume that the total value change per user per epoch can be represented with 3 bytes or less in the format of $x \cdot y \times 10^z$, a value $\leq 256^2 \cdot 10^{256}$, then we can see that a smaller data size than state-of-the-art rollups from Section 3.3.2 will be achieved if the target use case's systems' variable, average TX per user per epoch, is $\geq 1$ or $\geq 1.53$ if compared to AGC from Section 5.1. However, the more users that send or receive multiple TXs per epoch, the more this proposed value compression will reduce the number of messages and the data size of these messages.

Finally, this value batching can be done with zero-knowledge proofs as part of a coin mixing algorithm to increase security by obfuscating who is transacting with who, what, and when. However, as token tracking is done by a rollup source TX locally on the shards, not by the rollup, obfuscating who is transacting will need additional steps.

Value Batching Procedure

Algorithm 3 shows how each epoch a shard would process a set of valid TXs from Algorithm 1 for value batching. The TXs the rollup SC receives are processed, accumulating a value for each user. Towards the end of the epoch, these, as well as the IDs and opcodes from Algorithm 1, are entered into the rollup as a key pair in the format; user opcode + user ID: value,..., before the rollup is completed in line with Algorithm 2. By doing so, we reduce the number and size of messages being sent cross-shard, and the amount of reduction being determined by the user's systems' variable, average TX per user per epoch, at a computational cost of $\mathcal{O}(N)$, where $N$ is the number of TX received at a shard during an epoch, in order to achieve this compression.

---

**Algorithm 3** Value batching.

---

**Input**: $\mathcal{T}$, transactions arriving at a shards' rollup SC and,
$\mathcal{KP} = T/F$ depending on if the keypair or ordered list output is desired.
**Output**: $V$, an ordered list of the total value change per user or
$U$:$V$, User: Total Value change key pairs.

1: $V \leftarrow []$, a key-value storage (key: address, value: cumulative value)
2: $U \leftarrow \{\}$, a user list
3: **for** $TX$ in $\mathcal{T}$ **do**
4:     **if** $TX[sender]$ not in $U$ **then**
5:         $U.append(TX[sender])$
6:     **end if**
7:     **if** $TX[recipient]$ not in $U$ **then**
8:         $U.append(TX[recipient])$
9:     **end if**
10:     $V[TX[sender]] += TX[sender].value$
11:     $V[TX[recipient]] += TX[recipient].value$
12:     **if** $\mathcal{KP} == T$ **then return** the dictionary of User ID and Value key pairs; $U_0 : V_0, \ldots, U_n : V_n$
13:         **elsereturn** the order list of values; $V_0, \ldots, V_n$
14:     **end if**
15: **end for**

---

*5.4. Omission of the Radix IDs*

Our last proposal is to omit the radix IDs from TXs in the rollup body when most accounts' balances change in an epoch or a majority of a shard's accounts are involved in the same rollup. The idea is that, if all accounts' values change, then we can just retain the list of values instead of the pairs of accounts and values. We can do this for two reasons: Firstly, a section of the Merkle tree, which corresponds to the sharding committee, can be treated as an ordered list. Secondly, our proposed ID format is the Shard ID $+, 0, \ldots, k$. Therefore, we can modify Algorithm 3 to output an ordered list whose indices, a value between 0 and $k$, can be used in place of an ID. However, this would only offer compression if the majority of a shard's users are sending a single rollup to another shard as any non-participant user would need to be replaced with a 1-byte null opcode.

With this use of a 1-byte null opcode instead of a 3-byte long value, we can see that, if more than about 80% of a shard's accounts all participate in a rollup, it will offer data size compression. As such, it is in these user cases that we suggest using this proposal.

5.4.1. GVCRO's Assumptions

Therefore, the performance of GVCRO is dependent on the assumptions that: (i) Any user can be easily identified from the index of the fixed order list of the sharding committee taken from the Merkle tree. (ii) The number of non-participants is low; therefore, the ratio of null opcodes to values is in favor of the values; therefore, sending ID and value key pairs would result in larger data sizes.

5.4.2. Potential Use Cases for GVCRO

There exist many potential applications in both public and private Ethereum where sending a rollup involving the majority of accounts on a shard may occur, such as updating the coordinating layer at the end of an epoch, or, a single main supplier on another shard, or, using the coordinating layer as a hub to relay rollups. However, these conditions are more likely to occur in private Ethereum as we discussed in Section 1, where we talked about how private Ethereum could be deployed for large-scale systems, such as an EMS, and its typical conditions. For an EMS, these conditions would be in line with the assumptions of multiple TX per user per epoch, and more than 80% of a shard sending a rollup to the same shard that epoch. These conditions are due to the high P2P trade volume, the presence

of large suppliers and consumers, and the need to update the main system state on the coordinating layer each epoch.

## 6. Performance Evaluation

*6.1. Parameters for Performance Evaluation*

6.1.1. Evaluation Categories for Performance Evaluation

In this section, we will theoretically analyze how many TXs per block (TPB) or user states updated per block (USUPB) can be achieved with each of our proposed compressions. We do not use TPS as the metric to compare our proposals as our proposals only compress TX data sizes by changing the data format and do not change any part of the rollup algorithm involved in rollup validation, which is a determining factor for the TPS for rollups. Therefore, the TPS of our proposal will remain the same as the state-of-the-art [4]. However, if one wanted to get an idea of what TPS the verification algorithm must achieve to fill a block within the block time, one can divide TPB by the block interval to give a minimum TPS the verification algorithm will need to achieve.

6.1.2. System Parameters for Performance Evaluation

With regard to system parameters, we will be using those of public Ethereum, despite talking about and targeting compressions at private Ethereum. This is to ensure that our proposed compressions can be fairly evaluated; as such, we will not be modifying the system parameters to those of a typical private Ethereum blockchain, which would replace the consensus algorithm of public Ethereum 1.0, PoW, or Ethereum 2.0, PoS, with PoA as part of the privatization process and may change the block size, which is likely to result in large improvements in TPB, USUPB, and TPS. The parameters we will use in the evaluations are as follows: (i) A block gas limit of 15,000,000 gas as defined in [22]; (ii) A gas per byte for conventional TX of 16 *gas/byte*, which is an average based on analyzing historic TXs on Ethereum [4]; (iii) A gas per byte for rollups of 68 *gas/byte*, which again is an average based on analyzing historic rollups on Ethereum [34].

We can divide (i) the block gas limit by (ii) or (iii) multiplied by the TX byte length to give us an idea of either TPB or USUPB using the equation *block gas limit/gas per byte* × *TX byte length*.

We recognize that two values in (ii) and (iii) are averages based on historic analysis of the gas verification cost of the TX type in question, and as a result, they are subject to change. However, as mentioned above, we are not altering the verification process, so if we ran each compression on the same system state, we would be able to ensure that the value for gas per byte does not change.

6.1.3. Evaluation Parameters for Performance Evaluation

For each compression, we will evaluate both the maximum and minimum typical TX sizes to give ranges of TPB or USUPB which can be compared. For conventional TX, rollups, AC, and AGC, which all compress a TXs' data size, we will evaluate how many TXs one standard Ethereum block can hold. For AGVC and GVCRO, which instead record value change per user per epoch, we will evaluate how many users' states' can be updated in one standard Ethereum block. As explained in Section 5.3, the performance of AGVC and GVCRO is dependent on the variable "average TX per user per epoch" taken from the system where our proposal is deployed, making a conversion to TPB difficult. In Section 5.3, we did, however, recommend a minimum average TX per user per epoch for AGVC of 1.53 TX as this is the point when AGC will outperform AGVC.

6.1.4. Evaluation Process Used for Performance Evaluation

We begin our evaluation process by using the Ethereum protocols to create random addresses for both sender and recipient, random values for the value field, and a random value for gas. All of these random fields were constructed in line with Ethereum's parame-

ters and were validated on Ethereum's test net. We then used these dummy TX to validate our calculations to give us TPB and USUPB values for each compression.

In the next section, we will also use these dummy TX to evaluate our calculations for the data sizes of radix and Merkle trees with different $k$ values in order to evaluate the performance of our recommended $k = 256$ tree.

### 6.1.5. Message Parameters Used for Performance Evaluation

Next, we must address how we determine TX size for each TX type from Tables 1–6. Investigating the tables will show that the majority of fields in the TX have a variable size. Table 1 fields such as value can vary between 32 bytes, and a 1-byte opcode, to 0 bytes, and a 1-byte opcode. Table 2–6 all also have variable length value fields which are typically 2 or 3 bytes long with or without a 1-byte opcode. To account for this variable field size issue, the following calculations will use the worst-case maximum typical field sizes. However, for Figure 4, we will plot both the best and worst-case scenarios giving a range of performance.

**Table 6.** The data structure for a rollup using our proposed value compression and using the data structure of the TX in place of leaf IDs.

| TX FIELD | BYTES | OPCODE IN BYTES |
|---|---|---|
| HEADER | | |
| SHARD ID | 1 byte for up to 256 shards per layer & 1 byte per layer in a Merkle tree | 1 |
| ROLLUP SIZE | 1 byte per 256 bytes in rollup | 1 |
| GAS | 1 | 1 |
| POST-STATE-ROOT | 4 | 1 |
| SIGNATURE | 68 | 1 |
| BODY | | |
| VALUE for radix ID 0 | typically 2 or 3 | |
| … | … | |
| VALUE for radix ID $n$ | typically 2 or 3 | |

### 6.2. TPB

#### 6.2.1. Evaluation Parameters

Now that we have defined the variables, gas block limit, gas per byte, and the best and worst case TX byte lengths, we can now use these variables in the following calculations which show the worst-case scenarios, and we will plot these and the best-case scenarios in Figure 4 for both PB and USUPB for each compression method. In this figure, the bars show the average values while the upper and lower error bars show the best and worst-case scenarios based on whether all the source TX in a rollup are either the maximum typical size or minimum typical size outlined in those compression methods' TX structure table.

#### 6.2.2. Conventional TX

Let $g_x$ denote the total gas cost for TX type $x$. For instance, the conventional TX types' gas, $g_{\text{conv}}$, is derived as follows:

$$g_{\text{conv}} = 16(2 + 33 + 33 + 33 + 33 + 33 + 33 + 33 + 69 + 1). \tag{2}$$

Starting with conventional TX in Figure 4, we can see that this TX type achieves the lowest TPB when the conventional TX data structure is modified to be sent cross-shard.

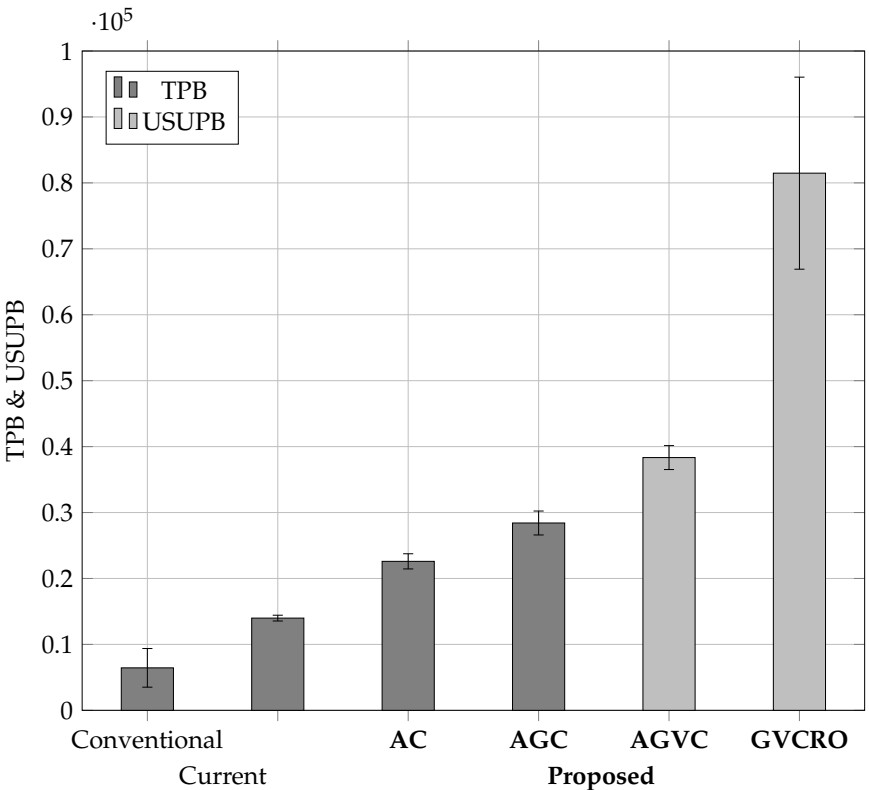

**Figure 4.** Comparison of the performance of each message and compression type using their best-case minimum message field size and worst-case maximum message field sizes to generate ranges and averages [4].

### 6.2.3. State-of-the-Art Rollups

In Equation (3), we change the TX type to rollup, necessitating a change in gas per byte in line with [4]. State-of-the-art rollups [4] significantly improve the TPB by compressing a message to about 16 bytes, depending on the length of the variable length fields in the rollup and the number of TX in the rollup as a portion of the 80-byte header must also be accounted for. It is this variability in field length that dictates the difference in performance between the best and worst-case scenarios, not messages per rollup as the rollup protocol heavily disincentivized users from sending a rollup when the header would form a non-negligible portion of the rollup. Instead, the rollup protocol keeps the message in the rollup pool until this negligible header size requirement is met.

$$g = 68(256(5 + 5 + 4 + 2) + 1 + 1 + 4 + 68)/256. \tag{3}$$

### 6.2.4. AC and AGC

The next Equations (4) and (5) show the worst-case performance of the compressions AC and AGC, respectively:

$$g_{\text{AC}} = 68(256(\mathbf{2} + \mathbf{2} + 4 + 2) + 1 + 1 + 4 + 68)/256. \tag{4}$$

$$g_{\text{AGC}} = 68(256(2 + 2 + 4) + \mathbf{2} + 1 + 1 + 4 + 68)/256. \tag{5}$$

Using Equation (4) and Table 3, we can see that there is a significant performance difference between state-of-the-art rollups [4] and AC. This is as AC reduces the sender and recipient field sizes from 5 to 2 bytes. This reduction in data size allows a huge increase in TPB, allowing us to pack 60% more cross-shard TXs into a block. AGC further improves on this by making the gas field common in the header, doubling the TPB of state-of-the-art rollups [4].

### 6.3. USUPB

6.3.1. Evaluation Parameters

In this section, we will now investigate AGVC and GVCRO using USUPB. As defined in Section 5.3, the use of the variable "average TX per user per epoch" makes converting to TPB difficult, necessitating a different comparison metric.

6.3.2. Assumptions

The compressions AGVC and GVCRO are targeted at, but not limited to, private Ethereum due to the utilization of the system state assumptions: (a) the average messages per user per epoch, and (b) the number of users, per shard, per epoch, that would send TXs to the same recipient shard, which we have used to allow us to more aggressively reduce the size and number of TXs included in rollups. Note that these assumptions may be satisfied in certain cases on public Ethereum; however, the conditions that would satisfy these assumptions are more common in private Ethereum.

6.3.3. Evaluation of AGVC and GVCRO

As described in Section 5.3, AGVC relies on compressing multiple TX into a net value change for both senders and recipients. Similarly, in Section 5.1.2 on GVCRO, we describe how both the senders' and recipients' radix IDs can be omitted when the system conditions allow us to only use the indices of the sharding committee in place of IDs to identify a users' value change.

Using the parameters outline in Section 6 and the field sizes from Tables 5 and 6, we obtain the Equations (6) and (7):

$$g_{\text{AGVC}} = 68(256(2+4) + 2 + 1 + 1 + 4 + 68)/256. \tag{6}$$

$$g_{\text{GVCRO}} = 68(256 * \mathbf{3} + 2 + 1 + 1 + 4 + 68)/256. \tag{7}$$

Placing these values into Figure 4, we can see that AGVC and GVCRO outperform all other compressions so long as the users' systems' variable "average TX per user per epoch" is $\geq 1.53$ as we derived in Section 5.3.

### 6.4. Impact of k-Ary Trees k Value on Performance

In Section 5.1, we proposed changing the *k*-ary tree type from radix to Merkle as it is only when a radix tree is saturated that all leaves fall on the same layer, a requirement for our proposal. Our proposal also requires us to be able to form groups based on arbitrary parameters that share a radix path, something we cannot do within the address-structured radix tree. Finally, for maximum address compression within the byte structure of the rollup message, we have recommended that the *k* value is increased from 16 to 256. This is to ensure all possible address slots are occupied.

There are several other benefits to changing the tree type and *k* value, aside from enabling the shorter addresses. Let us start with data size: using Figure 5, we can see the ratio of branch to leaf nodes for different values of *k*. Starting with a binary Merkle or radix tree, $k = 2$, we can see that only half of the nodes will be leaves storing data, and the remainder will be branches. Progressing to Ethereum's *k* value of $k = 2^4$, we can see that, for each increase by a power of 2, a halving in the number of branch nodes to leaf nodes has occurred. Thus, now only 1/16 of nodes in the tree are branches. Finally, we have our proposal of $k = 256$, where only 1/256 nodes in the tree are branches, further reducing the tree's total size. However, using the trend line, we can see that, if we were to increase the value for *k* any further, it would offer a diminishing return on *k*-ary tree data size compression.

The advantage of smaller data sizes is that it makes the tree easier to store, a key issue as storing data on a blockchain is expensive and results in many replicas of the data bloating the data size. Additionally, during our testing, we did not experience a noted increase in tree creation or verification times going from $k = 2$ to $k = 16$ and then $k = 256$.

However, much larger $k$ values will likely increase this when compared to a binary $k = 2$ Merkle tree with the same number of leaves. This increased computation to reprocess the $k$-ary Merkle Tree needs to be contrasted with the increase in TPS that a higher $k$ value would offer.

Finally, we must talk about the disadvantages of $k$-ary Merkle and radix trees, the proof size scales with the number of sister nodes such that $O(k \log_2(n))$ proofs are required versus a binary tree which has a proof size of $O(\log_2 n)$. To avoid this exponential growth in proof size, it is best to choose a lower $k$ value; however, this limits the amount of compression possible with our proposed address compression method. As a result, it is best to replace the algorithm used to generate proofs for the $k$-ary Merkle tree with one whose proof is a fixed length, $O(1)$, like KZG commitments [35]. We will address this proof size issue as well as the limitations of KZG proofs in a future work where we will show how we can modify KZG commitments to suit the rollup use case before then applying them to both reduce the data sizes of the proofs and to also reduce the computational costs of rollups for the prover, compiler, and verifier.

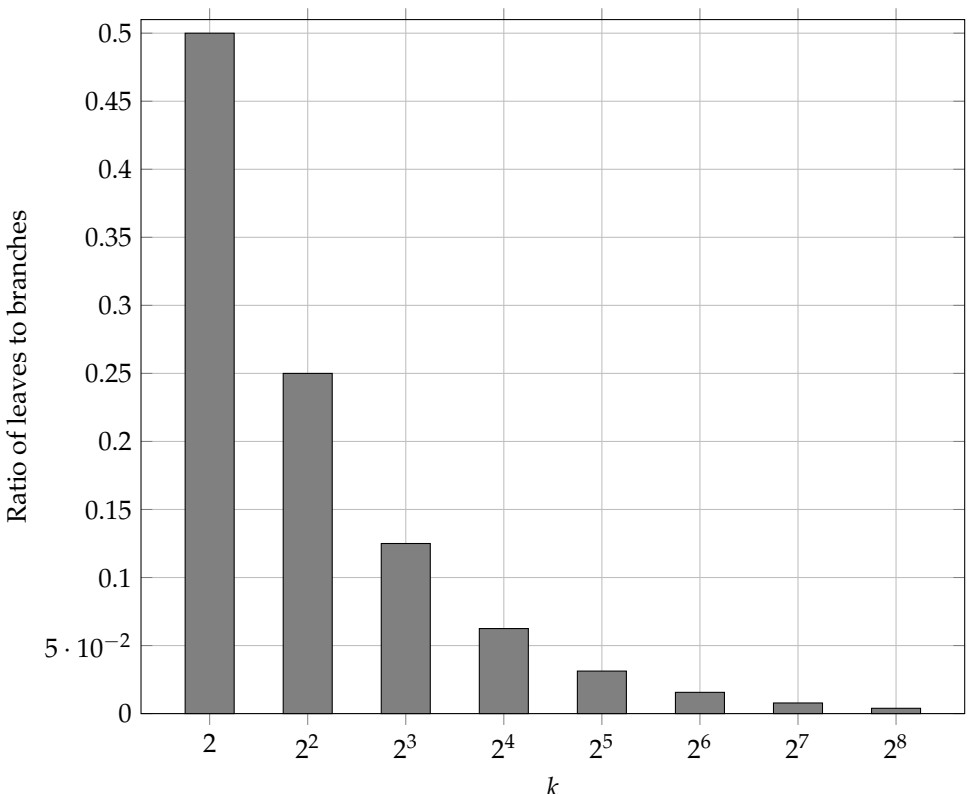

**Figure 5.** Ratio of branches to leaves for fully saturated $k$-ary trees with different K values $k = 2$, to $k = 2^8$. $k = 2^4$ and $2^8$ correspond to the current Ethereum and our proposal, respectively.

## 7. Conclusions

We have shown ways to increase the TPB of rollups in an account-based blockchain, such as Ethereum, through the use of $k$-Merkle trees whose data structure we have used to make discreet subsets within the Merkle tree corresponding to sharding committees. We have then used this data structure to enable a division of the radix path into two parts: one from the root to subset and another subset to leaf, thereby making the boundary node the shard ID where the shard state root is located. We have also removed any redundancies and compressed any repetitions, enabling an almost $2\times$ increase in TPB for rollups from $\approx$13,000 TPB to $\approx$30,000 TPB. Next, we went on to show how, for specific use cases, more common to private than public Ethereum, we can use TX batching techniques to effectively compress multiple TX into one state change per user. This change in rollup structure enables a scalar increase in performance in line with the average TX per user per epoch.

Finally, we progressed this proposal, removing further redundancies to create an approach that utilizes the data structure indices in the place of IDs. This further improved the number of user state changes we can compress into one block. We concluded this paper with a brief discussion on *k*-ary trees and their respective advantages and disadvantages, including smaller data sizes at the expense of larger proof sizes. We stipulated that KZG commitments, with their fixed $\mathcal{O}(1)$ proof size, can be used to resolve the proof size issue of *k*-ary trees by replacing the proof type used in *k*-ary trees and in Zero-Knowledge rollups with KZG commitments. However, to do so, the computational performance of KZG commitments needs to be improved, particularly when aggregating proofs. This is because a key step in rollup or *k*-ary tree generation is the generation of the proofs. To reduce the computational requirement, source KZG proofs can be aggregated; however, this currently poses a bottleneck that we believe can be much improved. We will show how this can be achieved in our next work.

**Author Contributions:** Conceptualization, methodology, validation, writing—original draft preparation, A.K.; writing—review and editing, supervision, K.T., A.I. and S.T. All authors have read and agreed to the published version of the manuscript.

**Funding:** This research received no external funding.

**Conflicts of Interest:** The authors declare no conflict of interest.

## Abbreviations

The following abbreviations are used in this manuscript:

| | |
|---|---|
| TPS | Transactions per Second |
| TPB | Transactions per Block |
| USUPB | User States Updated per Block |
| TX | Transactions |
| AC | Address Compression |
| AGC | Address and Gas Compression |
| AGVC | Address, Gas, and Value Compression |
| GVCRO | Gas and Value Compression no Radix ID |
| EMS | Energy Management System (for an electrical grid) |
| KZG | Kate, Zaverucha, and Goldberg's constant-sized polynomial commitments |
| PoW | Proof of Work |
| PoS | Proof of Share |
| PoA | Proof of Authority |
| SC | Smart Contract |
| EVM | Ethereum Virtual Machine |

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
