# Peer review of "Scaling Ethereum 2.0s Cross-Shard Transactions with Refined Data Structures"

_cryptography, doi:10.3390/cryptography6040057_

Round 1

Reviewer 1 Report

Relatively to the manuscript titled “Scaling Ethereum 2.0’s Cross-shard Transactions with Refined Data Structures, I believe that overall good work has been done in terms of quality and originality. The paper is well organized and written in sufficiently understandable English. Some minor issues have been found in the paper that should be solved to improve the journal readability:

-       The authors are suggested to move the related work section just after the introduction.

-       The authors are recommended to better introduce the novelties and potentialities of the proposed data structure compared to other commonly used in blockchain.

-       The acronyms should be clarified just at the first appearance. It isn’t needed to repeat the definition more times in the text (e.g., SC-smart contract).

-       The authors are advised to review the whole manuscript to correct typos and grammar issues.

-       The authors are suggested to review the bibliographic references to comply with the journal template (for further info, see: https://www.mdpi.com/journal/cryptography/instructions).

Author Response

We appreciate your time and kind comments which gave us valuable insights on how to improve the paper. The comments and replies are as follows. We have highlighted the modified manuscript in red.

Reviewer 1

1.1 Comments
Relatively to the manuscript titled “Scaling Ethereum 2.0’s Cross-shard Transactions with Refined Data Structures, I believe that overall good work has been done in terms of quality and originality. The paper is well organized and written in sufficiently understandable English. Some minor issues have been found in the paper that should be solved to improve the journal's readability:
1. The authors are suggested to move the related work section just after the introduction.
2. The authors are recommended to better introduce the novelties and potentialities of the proposed data structure compared to others commonly used in blockchain.
3. The acronyms should be clarified just at the first appearance. It isn’t needed to repeat the definition more times in the text (e.g., SC-smart contract).
4. The authors are advised to review the whole manuscript to correct typos and grammar issues.
5. The authors are suggested to review the bibliographic references to comply with the journal template (for further info, see: https://www.mdpi.com/journal/cryptography/instructions).

1.2
Replies
1. We have moved the related work section. 2. Our novelties against the state-of-the-art are our reduction in message size over the state-of-the-art with minimal impact on the system ensuring there are no performance penalties. We have improved the description of the novelties in Section 2, Related Work.
3. The acronym, SC, was defined twice in the introduction, thank you for spotting that oversight. We have checked the other acronyms, but no other cases were found in the paper.
4. We have reviewed and corrected the typos and grammatical errors as much as possible.
5. Thank you for highlighting this issue. We have reviewed the bibliographic
references and found that some URLs failed to properly import. References [4], [22], [31], and [33] have been corrected.

Reviewer 2 Report

Authors propose a series of novel data structures for the compiling of cross-shard TXs sent using rollups for both public and private Ethereum which will mitigate the scaling issue in a sharded blockchain that utilizes rollups for cross-shard communication.

  1. The authors should add some more background and motivation in the introduction.
  2. Contributions at the end of the introduction section should be improved.
  3. The summary at the end of the literature review should be focused on the limitations of related work.
  4. The conclusion is also very short. Extending this section to include a detailed plan for future work soon.
  5. What are the asymptotic bounds of the model when evaluated under different scenarios(at least one)?

Some recent relevant references can be cited in the manuscript

Zhang, Jingyu, Siqi Zhong, Jin Wang, Xiaofeng Yu, and Osama Alfarraj. "A storage optimization scheme for blockchain transaction databases." Computer Systems Science and Engineering 36, no. 3 (2021): 521-535.

Wang, Jin, Boyang Wei, Jingyu Zhang, Xiaofeng Yu, and Pradip Kumar Sharma. "An optimized transaction verification method for trustworthy blockchain-enabled IIoT." Ad Hoc Networks 119 (2021): 102526.

Xu, Zisang, Wei Liang, Kuan-Ching Li, Jianbo Xu, and Hai Jin. "A blockchain-based roadside unit-assisted authentication and key agreement protocol for internet of vehicles." Journal of Parallel and Distributed Computing 149 (2021): 29-39.

Author Response

We appreciate your time and kind comments which gave us valuable insights
on how to improve the paper. The comments and replies are as follows. We
have highlighted the modified manuscript in red.

Reviewer 2
2.1 Comments
Authors propose a series of novel data structures for the compiling of cross-shard TXs sent using rollups for both public and private Ethereum which will mitigate the scaling issue in a sharded blockchain that utilizes rollups for cross-shard communication.
1. The authors should add some more background and motivation in the introduction.
2. Contributions at the end of the introduction section should be improved.
3. The summary at the end of the literature review should be focused on the limitations of related work.
4. The conclusion is also very short. Extending this section to include a detailed plan for future work soon.
5. What are the asymptotic bounds of the model when evaluated under different scenarios(at least one)?
6. Some recent relevant references can be cited in the manuscript
Zhang, Jingyu, Siqi Zhong, Jin Wang, Xiaofeng Yu, and Osama Alfarraj. ``A storage optimization scheme for blockchain transaction databases.'' Computer Systems Science and Engineering 36, no. 3 (2021): 521-535.
Wang, Jin, Boyang Wei, Jingyu Zhang, Xiaofeng Yu, and Pradip Kumar Sharma. ``An optimized transaction verification method for trustworthy blockchain-enabled IIoT.'' Ad Hoc Networks 119 (2021): 102526.
Xu, Zisang, Wei Liang, Kuan-Ching Li, Jianbo Xu, and Hai Jin. ``A blockchain-based roadside unit-assisted authentication and key agreement protocol for internet of vehicles.'' Journal of Parallel and Distributed Computing 149 (2021): 29-39.
2.2 Replies
1. We have added some more background and motivation to the introduction.
2. We have added detailed contributions at the end of the introduction.
3. We have refined the summary at the end of the literature review to clearly convey the limitations of the related works.
4. We have added the possible future work in the conclusion.
5. We have actually evaluated three different scenarios, namely; best, worst, and conservative cases in Section 6. To recap the scenarios stated there, the best case is where we assume minimal message sizes. In contrast, the worst case is where we assume maximal message sizes Finally, the conservative case is mid-ground between these with a mix of message sizes. 6. We have added relevant papers from the suggested papers to the related work section.

Reviewer 3 Report

The article presents improved encoding and compression schemes that will enhance transaction-per-block in the Ethereum blockchain. 

The presentation is sound and easy to follow.

The article makes a number of assumptions that are not explicitly stated in the article and are not taken into account in the calculations. 

The main assumption is that the sharding schema will remain fixed and the sharding committees will remain fixed. If this is not the case, the information about the sharding needs to be included in the blocks too, to adjust for any changes to the sharding algorithms.

The second assumption is based on the nature and frequency of transactions between specific users within and outside of the shards. The worst-case scenario and the best-case scenario are not entirely well explained, and the conservative count should be taken for comparisons. 

Finally, the paper has been written before Ethreum 2.0 merge, and, it is necessary for the authors to update the paper to reflect the current state of the art in Ethereum 2.0 based on the actual chain. 

I consider those changes to be necessary, but relatively easy to implement in the article. 

Author Response

We appreciate your time and kind comments which gave us valuable insights on how to improve the paper. The comments and replies are as follows. We have highlighted the modified manuscript in red.

Reviewer 3
3.1 Comments
The article presents improved encoding and compression schemes that will enhance transaction-per-block in the Ethereum blockchain. The presentation is sound and easy to follow.
1. The article makes a number of assumptions that are not explicitly stated in the article and are not taken into account in the calculations.
2. The main assumption is that the sharding schema will remain fixed and the sharding committees will remain fixed. If this is not the case, the
information about the sharding needs to be included in the blocks too, to adjust for any changes to the sharding algorithms.
3. The second assumption is based on the nature and frequency of transactions between specific users within and outside of the shards. The worst-case scenario and the best-case scenario are not entirely well explained, and the conservative count should be taken for comparisons.
4. Finally, the paper has been written before Ethereum 2.0 merge, and, it is necessary for the authors to update the paper to reflect the current state of the art in Ethereum 2.0 based on the actual chain. I consider those changes to be necessary, but relatively easy to implement in the article.
3.2 Replies
1. We have added a subsection in Section 4 for the assumptions and high-lighted them in the introduction.
2. Actually, our assumption is that the sharding schema will dynamically change as in Ethereum’s specification [2, 1]. If the sharding schema will be fixed, it would not allow for the onboarding of new users or the updating of existing users. We have refined Sections 5.1 and 5.1.1 to more cleary state that the proposal is for a dynamic schema with dynamic sharding committees that do not need to remain fixed. We have also modified Sections 3.2.1, 3.2.2, 3.2.3, and 3.3.1 on the information about sharding and what is included in the blocks.
3. Thank you for pointing this out. As also replied to Reviewer 2’s fifth comment, summarising the scenario parameters from Section 6 and our prior response, we have the best-case scenario where we assume minimal message sizes. In contrast, the worst case is where we assume maximal message sizes. We used a conservative average for the evaluation. 4. Thank you for the suggestion. As you know, Ethereum is undergoing a multi-phase upgrade, part (1) merging PoW and PoS chains into a single PoS chain, has been completed, part (2) rolling out sharding and then part (3) native inter-shard rollups are yet to be implemented. We clarified the explanation in Section 3. Preliminaries, subsection 3.3. Ethereum 2.0, to explain how the first step of this upgrade affects our proposed message format, which is that it may affect the gas price, however, it will affect the gas price equally for all schemes, SOA, and conventional TX types, as the price is set by EIPS. As such using a historic average for comparison reasons, as part of one of our two comparison methods, does not affect the outcome which we have explained in Section 6.1. Parameters for performance evaluation, subsection 6.1.2. System parameters for performance evaluation.

References
[1] Vitalik Buterin. Eth2 shard chain simplification proposal, 2019. Last accessed 30 October 2022.
[2] Vitalik Buterin. Ethereum Whitepaper, 10 2021. Last accessed 30 October
2022.
